# Spatiotemporal changes in Netrin/Dscam1 signaling dictate axonal projection direction in *Drosophila* small ventral lateral clock neurons

**Jingjing Liu[1], Yuedong Wang[1], Xian Liu[1], Junhai Han[1,2]\*, Yao Tian[1]\***

[1]School of Life Science and Technology, the Key Laboratory of Developmental Genes and Human Disease, Southeast University, Nanjing, China; [2]Co-innovation Center of Neuroregeneration, Nantong University, Nantong, China

**Abstract** Axon projection is a spatial- and temporal-specific process in which the growth cone receives environmental signals guiding axons to their final destination. However, the mechanisms underlying changes in axonal projection direction without well-defined landmarks remain elusive. Here, we present evidence showcasing the dynamic nature of axonal projections in *Drosophila*'s small ventral lateral clock neurons (s-LNvs). Our findings reveal that these axons undergo an initial vertical projection in the early larval stage, followed by a subsequent transition to a horizontal projection in the early-to-mid third instar larvae. The vertical projection of s-LNv axons correlates with mushroom body calyx expansion, while the s-LNv-expressed Down syndrome cell adhesion molecule (Dscam1) interacts with Netrins to regulate the horizontal projection. During a specific temporal window, locally newborn dorsal clock neurons secrete Netrins, facilitating the transition of axonal projection direction in s-LNvs. Our study establishes a compelling in vivo model to probe the mechanisms of axonal projection direction switching in the absence of clear landmarks. These findings underscore the significance of dynamic local microenvironments in the complementary regulation of axonal projection direction transitions.

**\*For correspondence:**
junhaihan@seu.edu.cn (JH);
yaotian@seu.edu.cn (YT)

**Competing interest:** The authors declare that no competing interests exist.

## eLife assessment

This study provides insights into the mechanism of axonal directional changes, utilizing the pacemaker neurons of the circadian clock, the sLNVs, as a model system. The data were collected and analysed using **solid** methodology, resulting in **valuable** data on the interplay of signalling pathways and the growth of the axon. The study holds potential interest for neurobiologists focusing on axonal growth and development.

## Introduction

During nervous system development, neurons extend axons to reach their targets in order to build functional neural circuits (*Bentley and Caudy, 1983*; *Klose and Bentley, 1989*; *Mann et al., 2002*; *Shi et al., 2023*). The growth cone, which is specialized at the tip of the extending axon, is critical for receiving multiple guidance signals from the external environment to guide axon projection (*Agi et al., 2024*; *Lowery and Van Vactor, 2009*). Guidance cues regulate cytoskeletal dynamics through growth cone-specific receptors, steering axons via attractive or repulsive signals (*Tessier-Lavigne and Goodman, 1996*; *Araújo and Tear, 2003*; *Koch et al., 2012*; *Kolodkin and Tessier-Lavigne, 2011*; *Stoeckli, 2018*; *Zang et al., 2021*). Over the last 30 years, several guidance cues have been identified

and divided into four classical families: Semaphorins (*Kolodkin et al., 1992*; *Luo et al., 1993*; *Pasterkamp, 2012*; *Raper, 2000*; *Winberg et al., 1998*), Ephrins (*Cheng et al., 1995*; *Kania and Klein, 2016*; *Sperry, 1963*; *Wilkinson, 2001*), Netrins (*Colamarino and Tessier-Lavigne, 1995*; *Hong et al., 1999*; *Kennedy et al., 1994*; *Meijers et al., 2020*; *Pan et al., 2023*; *Serafini et al., 1994*), and Slits (*Agi et al., 2024*; *Battye et al., 1999*; *Kidd et al., 1999*; *Seeger et al., 1993*; *Tessier-Lavigne and Goodman, 1996*). The emergence of other kinds of guidance cues, including morphogens (*Charron et al., 2003*; *Ciani et al., 2004*; *Colavita et al., 1998*; *Wen et al., 2007*; *Yoshikawa et al., 2003*), growth factors (*Barde et al., 1982*; *Berkemeier et al., 1991*; *Leibrock et al., 1989*; *Maisonpierre et al., 1990*; *Short et al., 2021*), and cell adhesion molecules (*Rader et al., 1996*; *Rader et al., 1993*; *Williams et al., 1994*; *Wong et al., 1995*), increases the complexity of axon pathfinding.

When going through a long-distance pathfinding process in the complex and highly dynamic in vivo environment, axons frequently undergo multiple changes in their projection directions (*Chitnis and Kuwada, 1991*; *Hidalgo and Brand, 1997*; *Hutter, 2003*; *Isbister et al., 1999*; *McConnell et al., 1989*). Crucially, intermediate targets play a pivotal role by providing vital guiding information that enables axons to transition their projection directions and embark upon the subsequent stages of their journey, ultimately leading them to their final destination (*Bartoe et al., 2006*; *de Ramon Francàs et al., 2017*; *Stoeckli, 2018*; *Timofeev et al., 2012*; *Timofeev et al., 2012*). Certain intermediate targets can respond to the growth cone's signaling to maintain a dynamic balance between attractive and repulsive forces (*Bron et al., 2007*; *Martins et al., 2022*). One of the well-studied intermediate target projection types is the midline cross model (*Bernhardt et al., 1992*; *Evans et al., 2015*; *Holley, 1982*). As a clear landmark, axons need to project to the midline first and then leave it immediately (*Bovolenta and Dodd, 1990*; *Moreno-Bravo et al., 2019*; *Pignata et al., 2016*; *Tulloch et al., 2019*). In *vertebrates*, the midline floor plate cells secrete long-range attractive signals to attract commissural axons toward them (*Kennedy et al., 1994*; *Serafini et al., 1996*). Once there, these cells emit signals of rejection to prevent excessive closeness of axons, thus aiding the axon in departing the floor plate to reach its intended destination (*Kidd et al., 1998*; *Long et al., 2004*; *Stoeckli et al., 1997*). Comparatively, the typical axonal projection process within the in vivo milieu undergoes multiple transitions in projection direction without a distinct landmark, such as the midline (*Garel and López-Bendito, 2014*; *Gezelius and López-Bendito, 2017*; *López-Bendito and Molnár, 2003*; *Molnár et al., 2012*). Nevertheless, the precise mechanisms underlying the alteration of axonal projection direction in the absence of distinct landmarks remain to be fully elucidated.

*Drosophila* small ventral lateral clock neurons (s-LNvs) exhibit the typical tangential projection pattern in the central brain (*Agrawal and Hardin, 2016*; *Gummadova et al., 2009*; *Hardin, 2017*; *Helfrich-Förster, 1997*). The s-LNv axons originate from the ventrolateral soma and project to the dorsolateral area of the brain, and subsequently undergo a axon projection direction shift, projecting horizontally toward the midline with a short dorsal extension (*Helfrich-Förster et al., 2007*). Several studies have reported the impact of various factors, including the Slit–Robo signal (*Oliva et al., 2016*), Unc5 (*Fernandez et al., 2020*), Lar (leukocyte-antigen-related) (*Agrawal and Hardin, 2016*), dfmr1 (*Okray et al., 2015*), dTip60 (*Pirooznia et al., 2012*), and Dscam1 long 3′ UTR (*Zhang et al., 2019*), on the axon outgrowth of s-LNvs in the adult flies' dorsal projection. However, the mechanism underlying the projection direction switch of s-LNv axons remains largely unknown. Here, we show that s-LNvs and newborn dorsal clock neurons (DNs) generate spatiotemporal-specific guidance cues to precisely regulate the transition of projection direction in s-LNv axons. This regulation uncovers unexpected interdependencies between axons and local microenvironment dynamics during the navigation process, enabling accurate control over axon projection.

## Results

### s-LNvs axons change their projection direction in the early-to-mid third instar larvae

The stereotypical trajectory of the s-LNvs axon projection can be succinctly characterized as an initial vertical extension originating from the ventrolateral brain, followed by a directional pivot at the dorsolateral protocerebrum, ultimately leading to a horizontal projection toward the midline. Immunoreactivity for pigment-dispersing factor (PDF) (*Cyran et al., 2005*), representing the pattern of LNv neurons, is initially detected in the brains of first-instar larvae 4–5 hr after larval hatching (ALH)

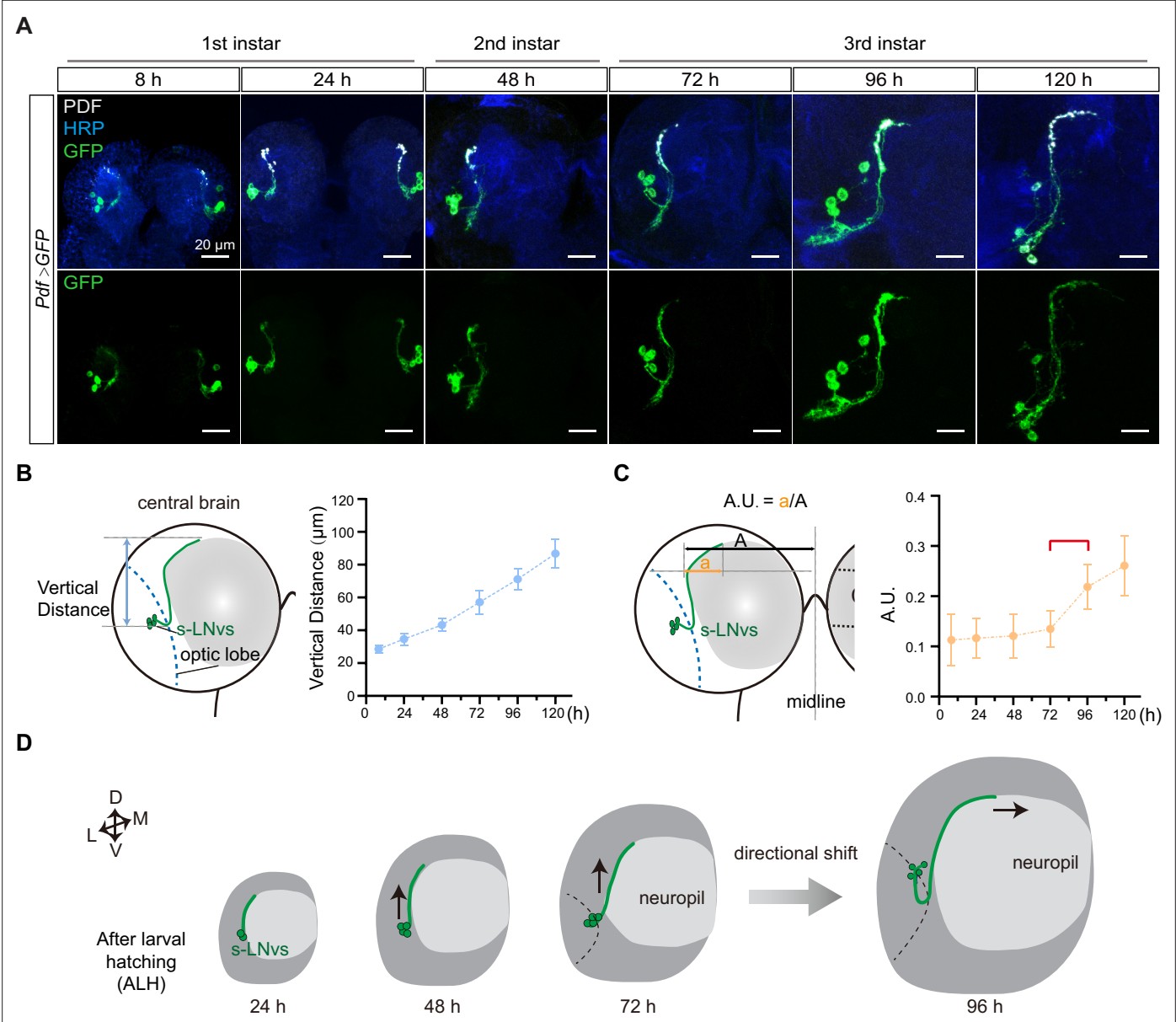

**Figure 1.** Small ventral lateral clock neurons (s-LNvs) axon projection dynamics during the larval stage. (**A**) Images showing the growth process of s-LNvs during larval development. Larval brains were stained with anti-PDF (white) and horseradish peroxidase (HRP, blue) antibodies. Different time points indicate the hours after larval hatching (ALH). (**B**) Left panel: schematic diagram illustrating the method used to measure the degree of vertical projection. One hemisphere of the larvae brain is depicted. Larval neuropil (gray), s-LNvs (green), and optic lobe (blue). Right panel: graphs showing the average vertical projection length at different developmental stages, presented as mean ± standard deviation (SD). (**C**) Left panel: schematic diagram illustrating the method used to measure the degree of horizontal projection. Right panel: graphs showing the average horizontal projection (A.U.: arbitrary unit) at different developmental stages, presented as mean ± SD. The red line segment indicates the stage at which the axonal projection undergoes a directional transition. (**D**) Schematic representation of s-LNvs vertical to horizontal projection directional shift during 72–96 hr ALH. One hemisphere of the larvae brain (gray), larval neuropil (white), and s-LNvs (green). For (**B, C**), 8 hr (n = 7), 24 hr (n = 10), 48 hr (n = 14), 72 hr (n = 16), 96 hr (n = 13), 120 hr (n = 16).

The online version of this article includes the following figure supplement(s) for figure 1:

**Figure supplement 1.** Small ventral lateral clock neuron (s-LNv) projection in the early larval stage.

(*Helfrich-Förster, 1997*). To elucidate the intricate axon pathfinding process during development, we employed a dual-copy *Pdf-GAL4* to drive the expression of double-copy UAS-mCD8::GFP, visualizing the s-LNvs at the larval stage (*Figure 1A*). We first tested whether *Pdf-Gal4* could effectively label s-LNv, and tracked the s-LNv projection in early larvae after embryo hatching. At 8 hr ALH, the green fluorescent protein (GFP) signal strongly co-localized with the PDF signal within the axons (*Figure 1—figure supplement 1*). This remarkable co-localization continued throughout the larval stage, particularly concentrated at the axon's terminal end (*Figure 1A*).

We then quantified the process of s-LNvs axonal projection. The vertical projection distance of the s-LNvs axon was ascertained by measuring the vertical span between the lowest point of the optical lobe and the highest point of the axon's projection terminal. Statistical analyses revealed a steadfast linear increase in the vertical projection distance of the s-LNvs axon as larval development progressed (*Figure 1A, B*). To determine the horizontal projection distance of the s-LNvs axon, we applied the methodology employed by Olive for measuring adult flies (*Oliva et al., 2016*). A tangent line is drawn precisely at the pivotal juncture where the s-LNvs axon undergoes a shift in its projection direction. We define the arbitrary unit (A.U.) as the ratio between the horizontal distances from this point to both the end of the axon projection and the midline (*Figure 1A, C*). Remarkably, at the time points of 8, 24, 48, and 72 hr ALH, the A.U. value for s-LNvs axon projection in the horizontal direction remained relatively stable at approximately 0.13. However, at 96 and 120 hr ALH, a significant and striking increase in the A.U. value ensued, eventually reaching approximately 0.25 (*Figure 1C*). These findings suggest that the precise transition of s-LNv axon projection from vertical toward the horizontal direction occurs in the early-to-mid third instar larvae, specifically during 72–96 hr ALH (*Figure 1D*).

## Vertical projection of s-LNvs axons correlate with mushroom body calyx expansion

The axons of s-LNv project dorsally, coming in close proximity to the dendritic tree of the mushroom body (MB), specifically the calyx (*Figure 2A, C*). This spatial arrangement facilitates potential interactions between the s-LNvs axon and the MB. MB predominantly comprises 2500 intrinsic neurons, known as Kenyon Cells (KCs) (*Puñal et al., 2021*). The development of the MB is a dynamic process characterized by three primary types of KCs: α/β, α′/β′, and γ, each following distinct temporal schedules of birth. The γ cells emerge from the embryonic stage until early third-instar larval phase, while the α′/β′ cells are born during the latter half of larval life, and the α/β cells appear in the post-larval stage (*Lee et al., 1999*; *Puñal et al., 2021*). We noticed that s-LNv axons exhibit only vertical growth before early third-instar larval stage, which coincides with the emergence of γ KCs. Consequently, we have contemplated the potential correlation between the vertical projection of s-LNv axons and the growth of the MB.

Next, we closely monitored the spatial relationship between the calyx and the axon projections of the s-LNvs throughout different larval developmental stages (*Figure 2A*). We used *OK107-GAL4* and *Tab2-201Y-GAL4* to drive the expression of mCD8::GFP in all KC subtypes or specifically in γ KCs (*Pauls et al., 2010*), respectively, to visualize MB KCs. As development progressed, the calyx area defined by both GAL4 drivers exhibited a consistent expansion. Statistical analyses further confirmed that the calyx area increased proportionally with the advancement of developmental time, demonstrating a linear relationship (*Figure 2B, C*). Additionally, Pearson correlation analysis unveiled a robust positive correlation between the calyx area and the vertical projection distance of s-LNvs axon ($r = 0.9987$, $p < 0.0001$ for *OK107-GAL4*; $r = 0.9963$, $p = 0.0003$ for *Tab2-201Y-GAL4*) (*Figure 2B, C*). These compelling observations suggest that the MB calyx, and potentially specifically the calyx of γ KCs, exert significant influence on the vertical projection of the s-LNvs axon.

To validate our idea, we employed a straightforward approach to evaluate any changes in s-LNvs axon projection after ablating γ KCs using the essential *Drosophila* cell death activators Reaper (Rpr) and Head involution defective (Hid) driven by the *Tab2-201Y GAL4*. Initially, we confirmed the effectiveness of our approach. In the control group, the GFP signal in γ KCs remained unchanged, while the ablation group showed a significant reduction or complete absence of the signal (*Figure 2—figure supplement 1*). Subsequently, we examined the axon projection of s-LNvs under these conditions. In the non-ablated group, the vertical distance increased linearly during development, following the expected pattern (*Figure 2D, E*). However, in the ablation group, although the initial vertical projection of s-LNvs appeared normal, there was a noticeable decrease in vertical projection distance at

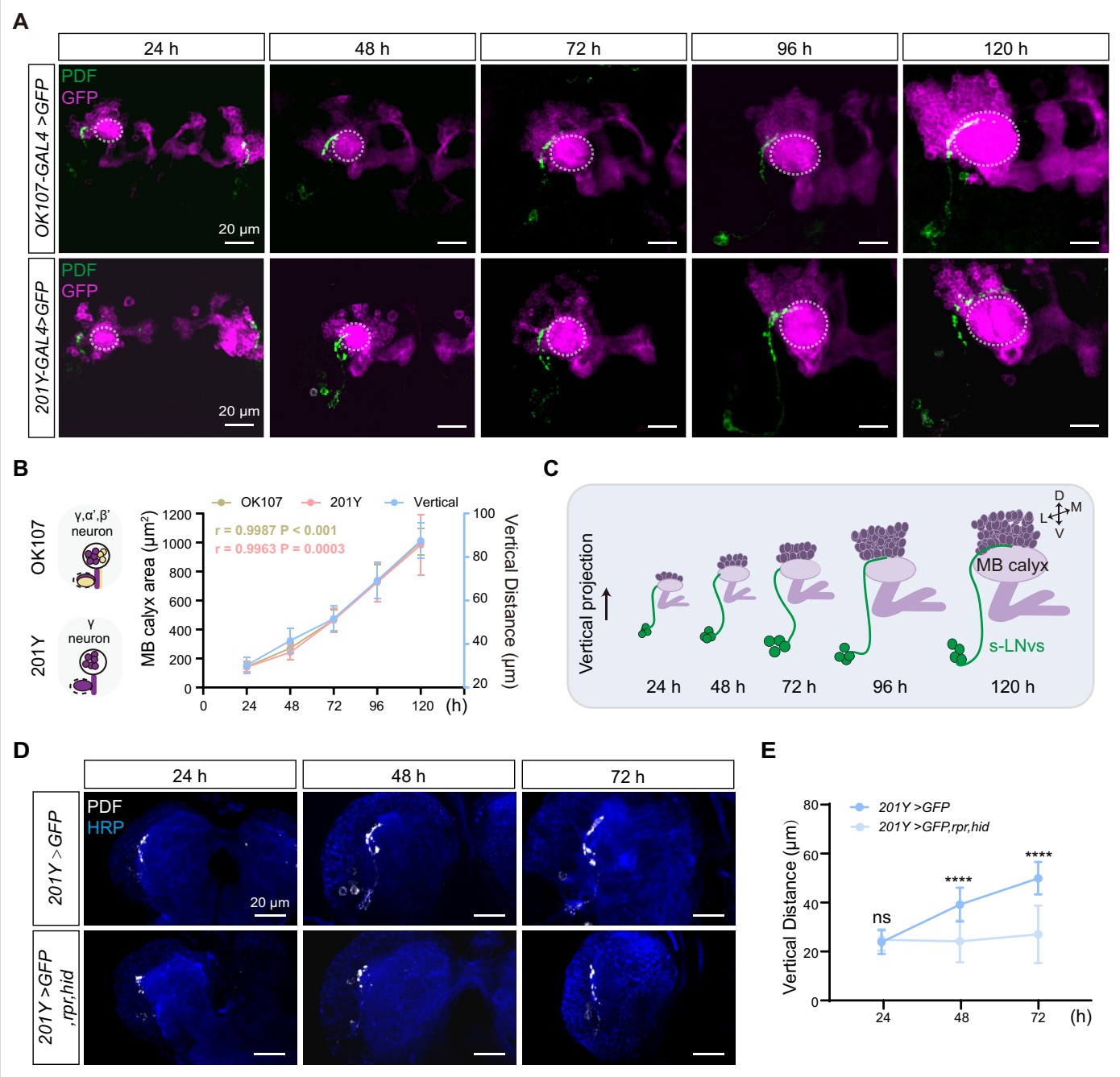

**Figure 2.** Development of vertical projection process alongside mushroom body (MB) growth. (**A**) Spatial position relationship between calyx and small ventral lateral clock neurons (s-LNvs). Top: *OK107-GAL4* labels the calyx of MB γ and α'β' neurons, while s-LNvs are labeled by the pigment-dispersing factor (PDF) antibody. Bottom: *Tab2–201Y-GAL4* labels calyx of MB γ neurons, with s-LNvs labeled by the PDF antibody. Larvae brains were stained with anti-PDF (green) and GFP (magenta) antibodies. Different time indicated hours after larval hatching (ALH). The white dotted line indicates the calyx region. (**B**) Graphs showing the MB calyx's area and vertical projection length averaged over different development stage are presented as mean ± standard deviation (SD). *OK107*: 24 hr (n = 10), 48 hr (n = 8), 72 hr (n = 12), 96 hr (n = 12), 120 hr (n = 9). Pearson's r = 0.9987, p < 0.0001. *Tab2-201Y*: 24 hr (n = 6), 48 hr (n = 12), 72 hr (n = 13), 96 hr (n = 13), 120 hr (n = 11). Pearson's r = 0.9963, p = 0.0003. (**C**) Schematic representation of complementary development of s-LNvs vertical projection and MB calyx. MB (purple), s-LNvs (green). (**D**) Images of MB ablation in the developing larval stages. Larvae brains were stained with anti-PDF (white) and HRP (blue) antibodies. Different time indicated hours ALH. (**E**) Quantification of vertical projection length in Control (*Tab2-201Y >GFP*) and Ablation (*Tab2-201Y >GFP,rpr,hid*) flies. Data are presented as mean ± SD. Control: 24 hr (n = 6), 48 hr (n = 14), 72 hr (n = 13), Ablation: 24 hr (n = 6), 48 hr (n = 12), 72 hr (n = 12). Two-tailed Student's *t* tests were used. ns, p > 0.05; ****p < 0.0001.

The online version of this article includes the following figure supplement(s) for figure 2:

**Figure supplement 1.** Validation of mushroom body (MB) ablation efficiency.

48 hr ALH. This vertical development trend ceased thereafter, with the vertical projection distance measuring only 20–30 µm at 72 hr ALH (*Figure 2D, E*). Importantly, in later larval stages, we observed that s-LNvs in certain larvae became undetectable (data not shown), suggesting the potential occurrence of apoptosis in these neurons following MB ablation. These results underscore a significant association between MB development, specifically the γ KCs, and the vertical projection dynamics of s-LNv axons.

## s-LNv-expressed Dscam1 mediate s-LNv axons horizontal projection

To decipher the signaling pathways governing s-LNvs axon horizontal projection, we performed a screening with a total of 285 targets by using the RNA-interfering (RNAi) approach (*Supplementary file 2*). To optimize efficiency, we chose *Clk856-GAL4*, which is expressed in s-LNv from the embryonic stage (*Figure 3—figure supplement 1A, B*), although it is also expressed in other clock neuron subsets (*Gummadova et al., 2009*). Significantly, the knockdown of Down syndrome cell adhesion molecule (Dscam1) resulted in a notable decrease in the horizontal projection distance of s-LNv axons in late third instar larvae and adults (*Figure 3—figure supplement 1C–F*). In adults, knockdown of Dscam1-L (including the exon 19 isoform) in s-LNvs results in the failure of axon terminals to form normally (*Zhang et al., 2019*). To validate the role of s-LNv-expressed Dscam1 in governing axon horizontal projection, we employed the *Pdf-GAL4*, which specifically targeted s-LNvs during the larval stage. At 96 hr ALH, vertical projection showed no significant changes in *Pdf-GAL4;UAS-Dscam1-RNAi* flies, but a noticeable deficit was observed in the horizontal projection of the s-LNvs axon (*Figure 3A–C*). These observations strongly indicate that Dscam1 specifically regulates the horizontal projection of s-LNvs axon.

We further checked s-LNvs axon projection in several well-defined *Dscam1* mutants, including *Dscam1$^{21}$*, *Dscam1$^{1}$*, and *Dscam1$^{05518}$* (*Hummel et al., 2003*; *Schmucker et al., 2000*). While all heterozygous mutants displayed normal horizontal axon projection in s-LNvs, trans-heterozygous mutants and homozygotes exhibited shortened horizontal axon projection (*Figure 3D, E*). Additionally, our immunostaining results indicated concentrated Dscam1 expression at the growth cone of s-LNv axons in third-instar larvae (*Figure 3F*). Taken together, these data demonstrate that Dscam1 controls the horizontal axon projection of s-LNvs in a cell-autonomous manner.

Dscam1 is a well-recognized cell adhesion molecule that plays a crucial role in axon guidance (*Chen et al., 2006*; *Hummel et al., 2003*; *Zhan et al., 2004*; *Zhang et al., 2019*). The receptors present on growth cones senses the extracellular axon guidance molecules to initiate the reorganization of the cellular cytoskeleton, leading to the facilitation of axonal projection. Convincingly, knockdown of the cytoskeletal molecules *tsr* (the *Drosophila* homolog of *cofilin*) (*Sudarsanam et al., 2020*) and *chic* (the *Drosophila* homolog of *profilin*) (*Shields et al., 2014*) in s-LNvs recaptured the horizontal axon projection deficits in *Dscam1* knockdown flies or *Dscam1* mutants (*Figure 3—figure supplement 1C, D*). Consistently, knockdown of *Dock* and *Pak*, the guidance receptor partners of Dscam1 (*Schmucker et al., 2000*), or *SH3PX1*, the critical linker between Dscam1 and the cytoskeleton (*Worby et al., 2001*), phenocopied *Dscam1* knockdown flies or *Dscam1* mutants (*Figure 3—figure supplement 2A, B*). Taken together, these findings provide solid evidence to support that Dscam1 signaling as a crucial regulator of s-LNvs axon horizontal projection (*Figure 3—figure supplement 2C*).

## Neuronal-derived Netrins act upstream of Dscam1 to govern the horizontal projection of s-LNvs axon

Two classical guidance molecules, Slit and Netrin, have been shown to specifically bind to the extracellular domain of Dscam1 (*Alavi et al., 2016*; *Andrews et al., 2008*). The *Drosophila* genome contains the sole *slit* gene and two *Netrin* genes: *Netrin-A* (*NetA*) and *Netrin-B* (*NetB*) (*Harris et al., 1996*; *Mitchell et al., 1996*). Notably, ubiquitous knockdown of both *NetA* and *NetB* (hereafter referred to as *Netrins*), rather than *slit*, resulted in the defective horizontal axon projection of s-LNvs. Convincingly, *NetA,NetB* double mutant (*Brankatschk and Dickson, 2006*), but not *NetA* or *NetB* single mutant, exhibited the defective horizontal axon projection of s-LNvs (*Figure 4A, B* and *Figure 4—figure supplement 1*). These results imply that two Netrin molecules function redundantly in regulating horizontal axon projection of s-LNvs.

To determine the source of Netrins, which are known to be secreted axon guidance molecules, we conducted experiments using pan-neuronal (*nsyb-GAL4*) and pan-glial (*repo-GAL4*) drivers

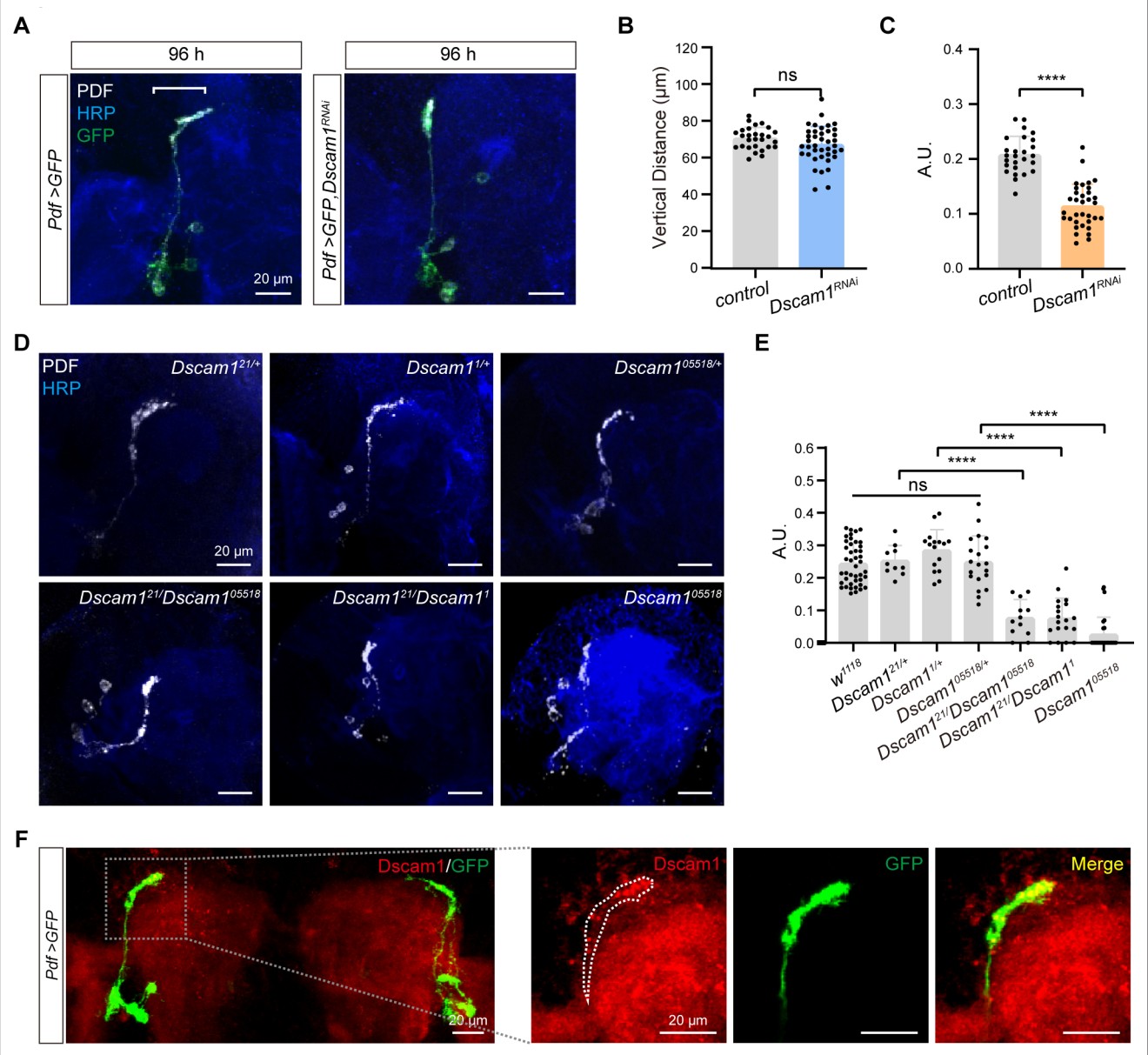

**Figure 3.** Critical role of Dscam1 in small ventral lateral clock neuron (s-LNv) axonal horizontal projection. (**A**) Images of s-LNvs in *Pdf >GFP* and *Pdf >GFP, Dscam1$^{RNAi}$* fly. White line segment represents the horizontal projection distance of s-LNvs. Larvae brains were stained with anti-PDF (white) and HRP (blue) antibodies at 96 hr after larval hatching (ALH). *Pdf >GFP* (n = 13), *Pdf >GFP, Dscam1$^{RNAi}$* (n = 15). (**B**) Quantification of vertical projection length in *Pdf >GFP* and *Pdf >GFP, Dscam1$^{RNAi}$* flies. Data are presented as mean ± standard deviation (SD). Two-tailed Student's *t* tests, ns, p > 0.05. (**C**) Quantification of horizontal A.U. in *Pdf >GFP* and *Pdf >GFP, Dscam1$^{RNAi}$* flies. Data are presented as mean ± SD. Two-tailed Student's *t* tests, ****p < 0.0001. (**D**) Images of *Dscam1* mutant s-LNvs projection phenotype. Larvae brains were collected at late third larvae. Heads were stained with anti-PDF (white) and HRP (blue) antibodies. (**E**) Quantification of horizontal A.U. in *Dscam1* mutant flies. Data are presented as mean ± SD. w$^{1118}$ (n = 22) *Dscam1$^{21}$/+* (n = 5), *Dscam1$^{1}$/+* (n = 8), *Dscam1$^{05518}$/+* (n = 10), *Dscam1$^{21}$/Dscam1$^{05518}$* (n = 6), *Dscam1$^{21}$/Dscam1$^{1}$* (n = 10), *Dscam1$^{05518}$* (n = 14). One-way analysis of variance (ANOVA) with Tukey's post hoc, ns, p > 0.05, ****p < 0.0001. (**F**) Endogenous Dscam1 co-localizes with the s-LNvs axon terminal. *Pdf >GFP* fly heads were collected at 120 hr ALH and stained with anti-Dscam1 (red). The white dotted line indicates the s-LNvs axon.

The online version of this article includes the following figure supplement(s) for figure 3:

**Figure supplement 1.** Knockdown of Dscam1 with *Clk856-GAL4* shortens small ventral lateral clock neuron (s-LNv) axon horizontal projection.

**Figure supplement 2.** Dscam1 and its downstream signaling pathway in small ventral lateral clock neuron (s-LNv) axonal horizontal projection.

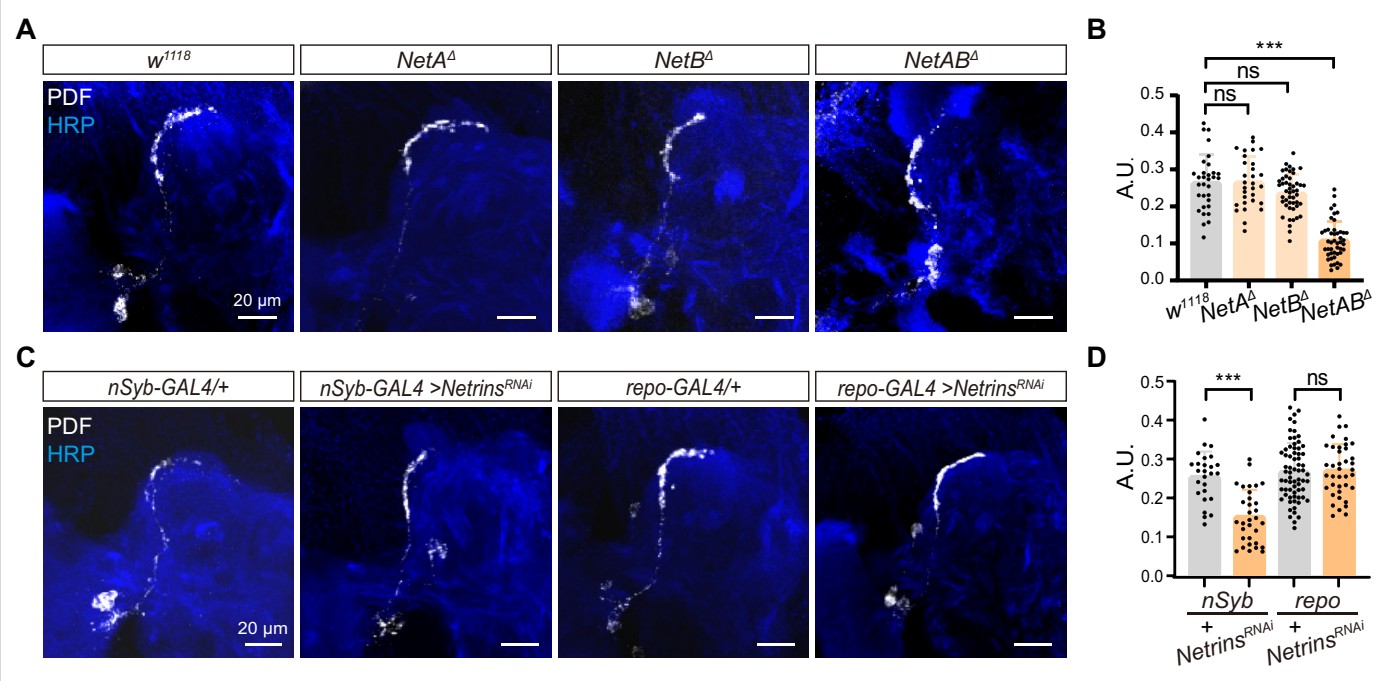

**Figure 4.** Neuron-derived Netrin guides small ventral lateral clock neurons (s-LNvs) horizontal projection. (**A**) Images of immunostained in $w^{1118}$, $NetA^\Delta$, $NetB^\Delta$, and $NetAB^\Delta$ fly. Larvae brains were collected at late third larvae. Heads were stained with anti-PDF (white) and HRP (blue) antibodies. (**B**) Quantification of horizontal A.U. $w^{1118}$, $NetA^\Delta$, $NetB^\Delta$, and $NetAB^\Delta$ fly. Data are presented as mean ± standard deviation (SD), $w^{1118}$ ($n = 17$), $NetA^\Delta$ ($n = 15$), $NetB^\Delta$ ($n = 24$), $NetAB^\Delta$ ($n = 23$). One-way analysis of variance (ANOVA) with Dunnett's post hoc, ns, $p > 0.05$, ***$p < 0.001$. (**C**) Images of immunostained in $nSyb$-GAL4 and $repo$-GAL4 knockdown Netrins. Larvae brains were collected at late third larvae heads were stained with anti-PDF (white) and HRP (blue) antibodies. (**D**) Quantification of horizontal A.U. in $nSyb$-GAL4 and $repo$-GAL4 knockdown Netrins fly. Data are presented as mean ± SD. $nSyb$-GAL4/+ ($n = 13$), $nSyb$-GAL4 >Netrins$^{RNAi}$ ($n = 16$), $repo$-GAL4/+ ($n = 34$), $repo$-GAL4 >Netrins$^{RNAi}$ ($n = 20$). Two-tailed Student's $t$ tests, ns, $p > 0.05$, ***$p < 0.001$.

The online version of this article includes the following figure supplement(s) for figure 4:

**Figure supplement 1.** Identifying the upstream signal of Dscam1.

to selectively knock down *Netrins* (**Figure 4C, D**). Strikingly, when *Netrins* were knocked down in neurons but not in glia, we observed severe defects in the horizontal axon projection of s-LNvs. These findings reveal that neuron-secreted Netrins serve as ligands for Dscam1, controlling the horizontal axon projection of s-LNvs.

## DN-secreted Netrins mediate the horizontal axon projection of s-LNvs

To further identify the source of Netrins, we focused on the neurons located in the dorsolateral protocerebrum, where the s-LNvs axons change their projection direction. We first excluded the possibility that MB-secreted Netrins mediate the horizontal axon projection of s-LNvs, as knockdown of *Netrins* with *OK107-GAL4* showed normal horizontal axon projection of s-LNvs (**Figure 5A, B**).

The DNs, a subset of clock neurons, also situate in the dorsolateral protocerebrum. DNs are categorized into three types, DN1, DN2, and DN3 (**Reinhard et al., 2022**), and previous studies have shown that both DN2 and DN1 form synaptic connections with s-LNvs axon terminals at the adult stage (**Schlichting et al., 2022**). Interestingly, knockdown of *Netrins* in a substantial portion of DN2, DN3, and DN1 (**Kaneko et al., 1997**) (*Per-GAL4, Pdf-GAL80;UAS-Netrins-RNAi*) resulted in the defective horizontal axon projection of s-LNvs (**Figure 5A, B**). To further validate the involvement of DNs in mediating the horizontal axon projection of s-LNvs, we conducted cell ablation experiments. Notably, the impairments in the horizontal axonal projection of s-LNvs due to DNs ablation were exclusively observed at 96 hr ALH, which is after the occurrence of horizontal projection (**Figure 5C, D**). In contrast, ablating crz$^+$ neurons, which occupy a similar location to DNs, had no significant effects on the horizontal axon projection of the s-LNvs (**Figure 5—figure supplement 1**). Moreover, by expressing NetB in DNs in *NetA,NetB* double mutants, we successfully restored the defective

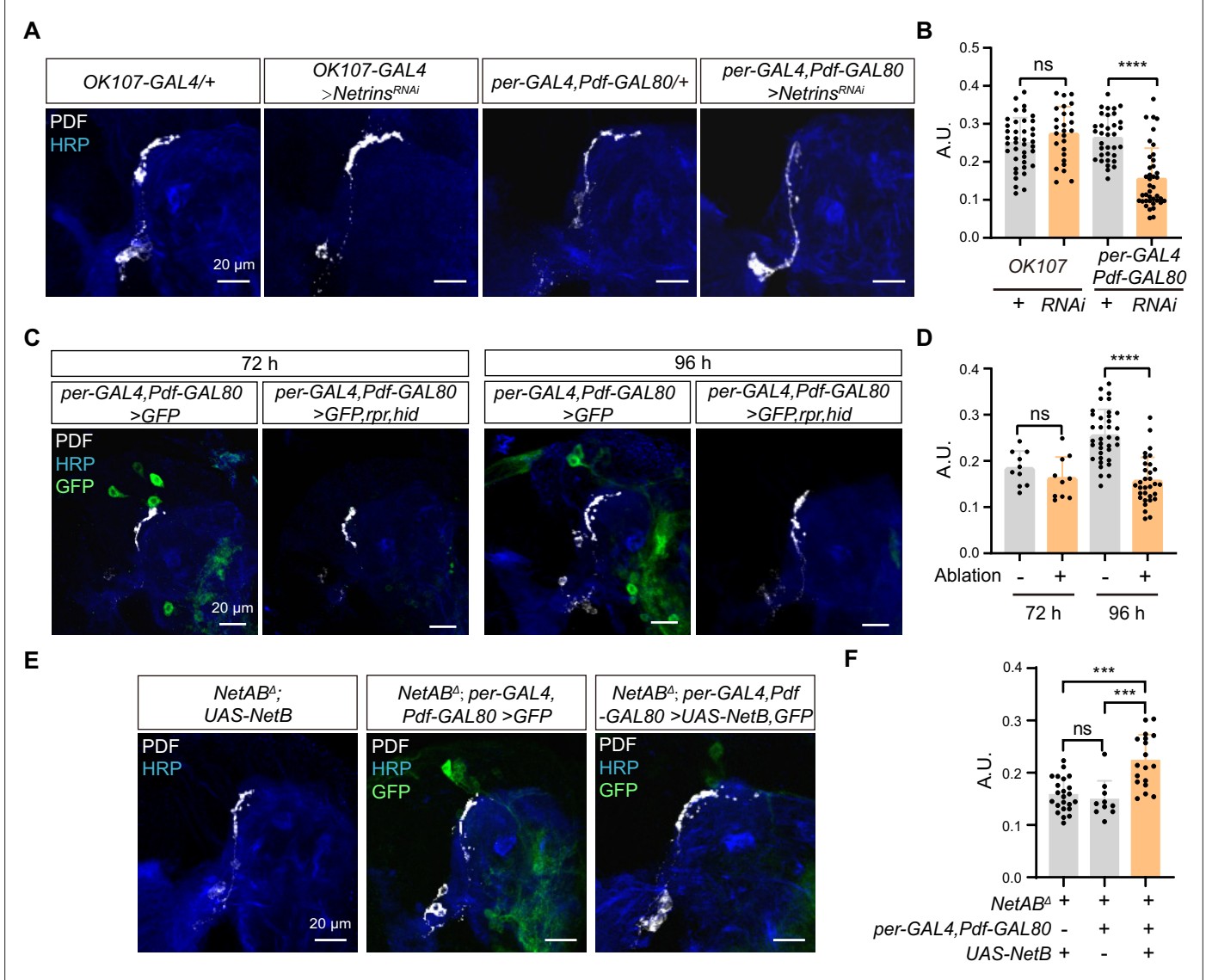

**Figure 5.** Netrin secreted by dorsal clock neurons (DNs) guides time-specific horizontal projection of small ventral lateral clock neurons (s-LNvs).
(**A**) Images of immunostained in *OK107-GAL4* and *per-GAL4, Pdf-GAL80* knockdown *Netrins*. Larvae brains were collected at late third larvae. Heads were stained with anti-PDF (white) and HRP (blue) antibodies. (**B**) Quantification of horizontal A.U. in *OK107-GAL4* and *per-GAL4, Pdf-GAL80* knockdown *Netrins* fly. Data are presented as mean ± standard deviation (SD). *OK107-GAL4/+* (*n* = 20), *OK107-GAL4>Netrins^RNAi^* (*n* = 13). *per-GAL4, Pdf-GAL80/+* (*n* = 18), *per-GAL4, Pdf-GAL80 >Netrins^RNAi^* (*n* = 16). Two-tailed Student's *t* tests for *OK107-GAL4* knockdown *Netrins*, ns, p > 0.05. Mann–Whitney test for *per-GAL4, Pdf-GAL80* knockdown *Netrins*, ****p < 0.0001. (**C**) Images of DN ablation in the developing larval stages. Larvae brains were stained with anti-PDF (white) and HRP (blue) antibodies. Different time indicated hours ALH. (**D**) Quantification of horizontal A.U. in Control (*per-GAL4, Pdf-GAL80 >GFP*) and Ablation (*per-GAL4, Pdf-GAL80 >GFP,rpr,hid*) flies. Data are presented as mean ± SD. Control: 72 hr (*n* = 5), 96 hr (*n* = 12), Ablation: 72 hr (*n* = 5), 96 hr (*n* = 10). Two-tailed Student's *t* tests were used to compare conditions. ns, p > 0.05, ****p < 0.0001. (**E**) Images of immunostained in *per-GAL4, Pdf-GAL80* overexpress *NetB* in *NetAB^Δ^*. Larvae brains were collected at late third larvae. Heads were stained with anti-PDF (white) and HRP (blue) antibodies. (**F**) Quantification of horizontal A.U. in *per-GAL4, Pdf-GAL80* overexpress *NetB* in *NetAB^Δ^*. Data are presented as mean ± SD. *NetAB^Δ^*, *UAS-NetB* (*n* = 12), *NetAB^Δ^, per-GAL4,Pdf-GAL80 /+* (*n* = 5), *NetAB^Δ^,per-GAL4,Pdf-GAL80 >UAS* NetB (*n* = 9). One-way analysis of variance (ANOVA) with Bonferroni post hoc, ns, p > 0.05, ***p < 0.001.

The online version of this article includes the following figure supplement(s) for figure 5:

**Figure supplement 1.** Ablation of dorsal clock neurons (DNs) leads to reduced small ventral lateral clock neurons (s-LNvs) horizontal projection.

horizontal axon projection of s-LNvs (*NetAB^Δ;Per-GAL4, Pdf-GAL80;UAS-NetB*) (**Figure 5E, F**). These results demonstrate that DNs secrete Netrins to guide the horizontal axon projection of s-LNvs.

## Newborn DNs secrete Netrins to regulate the horizontal axon projection of s-LNvs

Finally, we wonder which population of DNs secrete Netrins to regulate the horizontal axon projection of s-LNvs. Therefore, we monitored the number and location of DNs during the s-LNvs axon projection and found that the number of DNs significantly increased during this process. *Per-GAL4, Pdf-GAL80; UAS-mCD8::GFP* only labeled 4–5 cells at 48 hr ALH, and raise to 10–15 by 72 hr, and subsequently increased to approximately 25 by 96 hr ALH (**Figure 6A, B**). It is worth noting that the location of these newly formed DNs resides lateral to the transition point of axon projection direction of the s-LNvs, while maintaining a basic parallelism with the horizontal axon projection of the s-LNvs (**Figure 6A**). The sharp increase in the number of DNs coincides remarkably with the timing of the switch in axon projection direction of the s-LNvs.

Next, we asked whether these newly generated DNs expressed Netrins. To visualize the expression of endogenous NetB, we uesd the fly strains that insert either GFP or myc tag in *NetB* gene. At 72 hr ALH, we were unable to detect any NetB signals within the DNs. In contrast, at 96 hr ALH, we easily detected prominent NetB signals in approximately six to eight newborn DNs (**Figure 6C, D**, **Figure 6—figure supplement 1A**).

Previous studies have shown that DSCAM is involved in Netrin-1-mediated axonal attraction (**Andrews et al., 2008**; **Liu et al., 2009**; **Ly et al., 2008**). In addition, DSCAM also functions as a repulsive receptor, associating with Uncoordinated-5C (UNC5C) to mediate Netrin-1-induced axon growth cone collapse (**Fernandez et al., 2020**; **Purohit et al., 2012**). To dissect the role of Netrin signaling, we ectopically expressed Netrins in neurons marked by *R78G02-GAL4* (**Jenett et al., 2012**; **Suzuki et al., 2022**), located in front of the horizontal projection of s-LNv axons, closer to the midline (**Figure 6—figure supplement 1B**). Expression of either NetA or NetB in *R78G02-GAL4*-labeled neurons significantly suppressed the horizontal axon projection of s-LNvs (**Figure 6—figure supplement 1B, C**). Taken together, these findings reveal that newborn DNs secrete Netrins to orchestrate the transition of axon projection in s-LNvs from a vertical to a horizontal direction (**Figure 7**).

## Discussion

The mechanisms underlying axonal responses to intricate microenvironments and the precise development of axons within the brain have long remained enigmatic. In the past, researchers have identified a large number of axon guidance cues and receptors using different models in vivo and in vitro (**Stoeckli, 2018**). In addition, transient cell–cell interactions through intermediate targets play a crucial role in guiding axonal projection step by step toward its final destination (**Chao et al., 2009**; **Garel and Rubenstein, 2004**). The landmark midline, a crucial intermediate target, serves as a model in the majority of early studies exploring the directional transitions of axonal projections (**Evans and Bashaw, 2010**). However, so many neural projections do not cross the midline that it is challenging to understand how these axon projections are guided (**Goodhill, 2016**). In our study, we established an excellent model to investigate the mechanism of axonal projection direction switch by the *Drosophila* s-LNvs. We discovered a coordinated growth pattern between the vertical projection of s-LNvs and the MB calyx. Furthermore, the complementary interaction between Dscam1 expressed in s-LNvs and the emerging DN-secreted Netrins precisely modulates the transition of s-LNv axons from a vertical to a horizontal projection within a specific time window (**Figure 7**). These findings reveal the mechanism behind the transition of axonal projection direction, emphasizing the significance of developmental microenvironments in ensuring precise axon projection.

### The dependence of s-LNv vertical projection on MB calyx expansion

During our monitoring of the axonal projection of s-LNvs, we observed that the axon terminals reached the dorsolateral brain area at an early stage but continued their vertical growth. This phenomenon raises the question of what drives this growth. The MB, a sophisticated central hub within the fruit fly's brain, exhibits proximity to the projection of s-LNv axons in spatial arrangement. Upon careful examination, we found a positive correlation between the vertical length increase of s-LNv axons

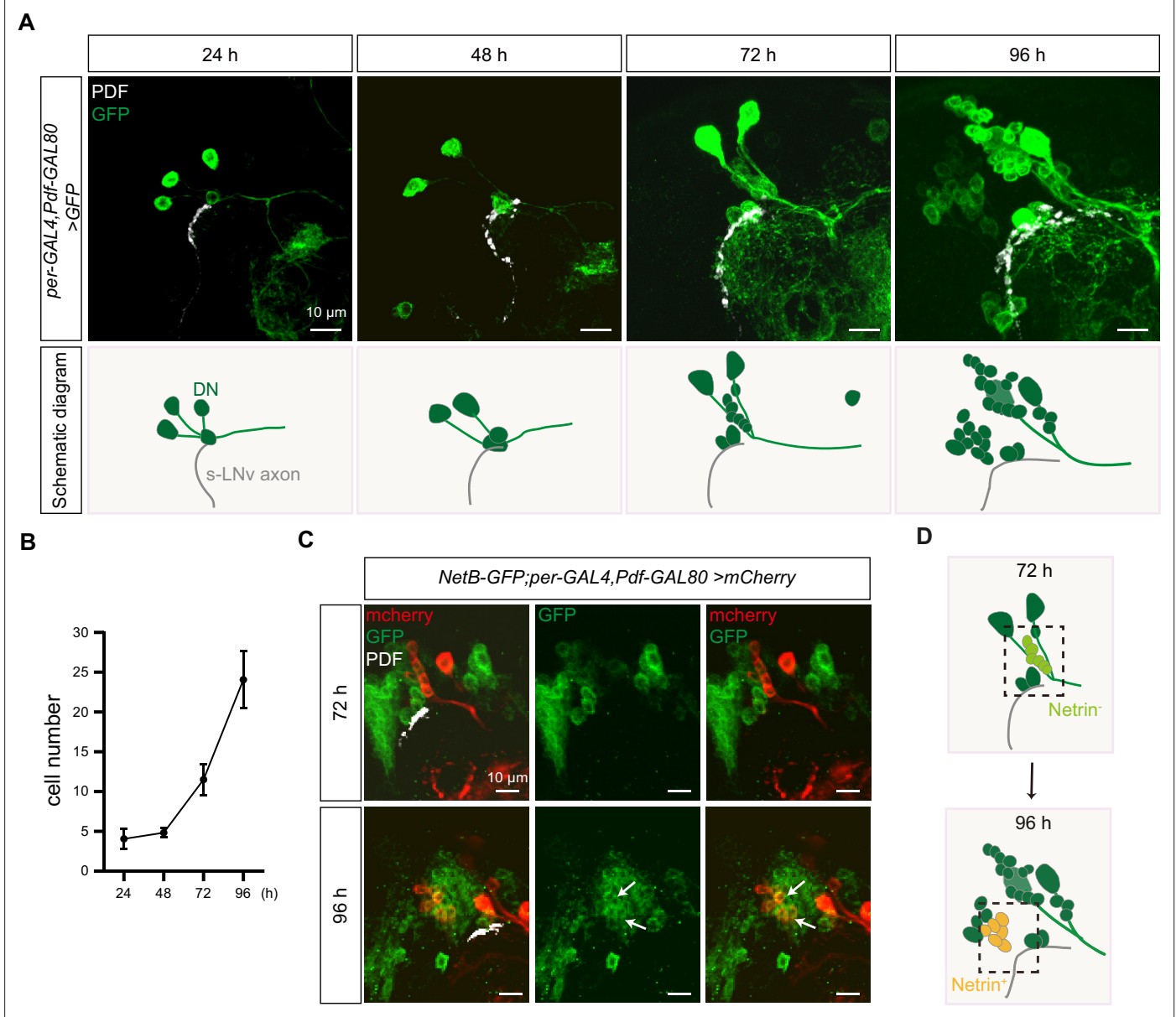

**Figure 6.** Dynamic changes in dorsal clock neurons (DNs) coordinated with small ventral lateral clock neurons (s-LNvs) axonal targeting. (**A**) Top: images of s-LNvs axon and DNs growth process in the developing larval stages. Larvae brains were stained with anti-PDF (white) and GFP (green) antibodies. Different time indicated hours after larval hatching (ALH) are shown. Bottom: schematic diagram of s-LNvs horizontal projection and the corresponding increase in the number of DN neurons. s-LNvs axon (gray), DN neurons (green). (**B**) Quantification of the number of DN neurons labeled by *per-GAL4, Pdf-GAL80* at different developmental stages. Data are presented as mean ± standard deviation (SD). (**C**) Images of newborn DN neurons were co-localized with *Netrin-B* at different developmental times. Larvae brains were stained with anti-GFP (green), mcherry (red), and PDF (white) antibodies. Different time indicated hours ALH. The white arrow demarcates the co-localization of red and green signals. (**D**) Schematic representation s-LNvs projection directional transition and the corresponding increase in the number of DN neurons. s-LNvs axon (gray), DN neurons (green), and newborn DN neurons (light green at 72 hr, orange at 96 hr).

The online version of this article includes the following figure supplement(s) for figure 6:

**Figure supplement 1.** Anterior Netrin ectopic expression reduces horizontal projection length.

and the growth of the MB calyx (*Figure 2*). Remarkably, when we selectively removed KC cells, we observed a striking effect on the axonal projection of s-LNvs. The s-LNv axonal projections either stalled at the initial stage or even completely disappeared (*Figure 2* and data not shown). Hence, the vertical projection of s-LNv axons is dependent on MB calyx expansion. Our hypothesis is supported by the findings of Helfrich-Förster who reported locomotor activity and circadian rhythm defects in

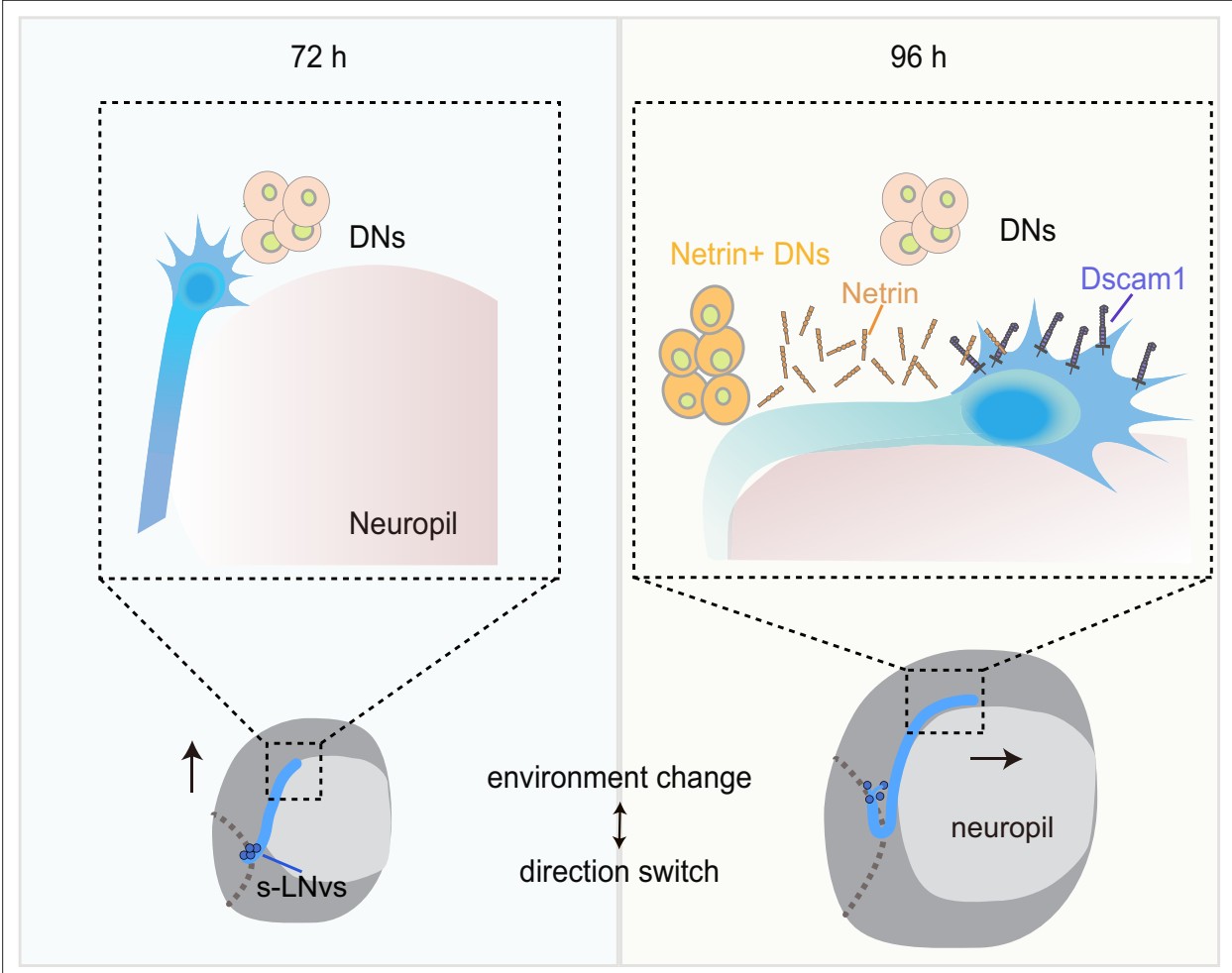

**Figure 7.** Dynamic cellular molecular environment during the small ventral lateral clock neuron (s-LNv) projection directional shift. Cartoon depicting the dynamic cellular molecular microenvironment during the axonal projection directional shift from vertical to horizontal projection in s-LNvs. 72 and 96 hr represent 72 and 96 hr after larval hatching, respectively. s-LNvs (blue), neuropile (pink), brain lobe (gray circle), dorsal clock neurons (DNs; orange and light orange circles), optic lobe (gray dotted line), Netrin (orange combination molding), and Dscam1 (purple).

some MB mutants (*Helfrich-Förster et al., 2002*), potentially attributed to the abnormality of s-LNvs axon. Moreover, it was observed that the MB mutants had minimal impact on the l-LNvs. This can be attributed to the fact that l-LNvs fibers emerge at a later stage during pupal development and are situated at a spatial distance from the MB.

Unfortunately, we did not identify any molecules that have a discernible impact on the vertical projection of s-LNv axons through screening. This indicates that the successful completion of the vertical projection process may play a pivotal role in determining the overall existence of s-LNv axonal projections. Furthermore, the dorsal projection of s-LNv axons intersects with the ventral projection of neurons such as DN1 in the dorsolateral region adjacent to the calyx (*Keene et al., 2011*), resembling the corpus callosum in mammals (*Fothergill et al., 2014*; *Hutchins et al., 2011*; *Keeble et al., 2006*; *Piper et al., 2009*; *Unni et al., 2012*). Thus, during early developmental stages, multiple guiding cues may redundantly function between the MB calyx and neighboring neural processes to ensure the smooth progression of neural development and establish a stable brain structure. However, to gain a more comprehensive understanding, it is worthwhile to explore whether the development of the MB calyx influences the projection of other neurons traversing the same territory and the guidance cues involved. Insights from existing single-cell sequencing data obtained from multiple stages of larval life (*Brunet Avalos et al., 2019*; *Corrales et al., 2022*), might offer some clues and indications.

## Netrin/Dscam signaling specifically controls the horizontal projection of s-LNv axons

In this paper, when we specifically knockdown *Dscam1* in s-LNvs, we observed a significant defect in the axonal horizontal projection. This suggests that Dscam1 autonomously regulates the horizontal axonal projection of s-LNvs (*Figure 3*). Our findings align with a recent study that reported abnormal horizontal axonal projection of s-LNv neurons in adult flies when Dscam1 mRNAs lack the long 3′ UTR (*Zhang et al., 2019*).

The extracellular domain of Dscam1 has been verified to be capable of binding to Netrin or Slit (*Alavi et al., 2016*). In *Drosophila*, at the midline of the embryonic central nervous system, Dscam1 forms a complex with Robo1 to receive the Slit signal and promote the growth of longitudinal axons (*Alavi et al., 2016*). In adult flies, Slit/Robo signaling restricts the medial growth of s-LNv axons (*Oliva et al., 2016*). However, we found that reducing *Netrins* levels, rather than *slit*, led to a decrease in the horizontal projection distance of s-LNv axons. Moreover, in the *NetAB$^\Delta$* double mutant, the extent of defects in the horizontal axonal projection of s-LNv neurons is similar to that observed when knocking down *Dscam1* in s-LNvs. Indeed, previous studies have provided evidence showcasing the widespread involvement of Dscam1 as a receptor for Netrin in mediating axon growth and pathfinding (*Liu et al., 2009*; *Ly et al., 2008*; *Matthews and Grueber, 2011*). When we disturbed the expression of Netrin signals in the axon targeting microenvironment, it was also sufficient to cause abnormal s-LNv projection (*Figure 6—figure supplement 1B, C*). Therefore, our findings suggest that Netrin, as a ligand for Dscam1, regulates the process of switching axon projection direction (*Figure 4*).

## The newborn DN-secreted Netrin coordinates with s-LNv-expressed Dscam1 to switch the projection direction of s-LNv axons

While the midline serves as a prominent landmark, it presents a daunting challenge to comprehend the guidance and directional transitions of axons in many neural projections that do not actually cross this central axis (*Evans and Bashaw, 2010*). Segmented patterns such as the limb segment traversed by Ti1 neuron projections or neural circuits formed in a layer- or column-specific manner also serve as intrinsic 'guideposts' (*Bentley and Caudy, 1983*; *Isbister et al., 1999*; *Kolodkin et al., 1992*), offering valuable insights into axonal pathfinding processes. How axonal projections without clear landmarks are guided? Following vertical projection, the s-LNvs growth cones remain within the dorsolateral area for at least 48 hr before initiating horizontal projection (*Figure 1*). This phenomenon aligns harmoniously with the outcomes observed in earlier studies that the axonal growth of other types of neurons slows down at the specific choice point, the midline (*Bak and Fraser, 2003*; *Godement et al., 1994*; *Li et al., 2021*), indicating that the dorsolateral area serves as an intermediate targets to facilitate the subsequent phase of s-LNvs projection journey.

Extracellular cues released by the final or intermediate targets play a crucial role in guiding axonal projection. The axon terminals of s-LNvs showed close spatial associations with the somas and processes of DNs. In this study, the knockdown of *Netrins* specifically in per$^+$,Pdf$^-$ neurons, but not in the MB, resulted in axon projection defects in s-LNvs (*Figure 5*). Furthermore, the ablation of DNs resulted in the inhibition of horizontal growth of s-LNvs axons (*Figure 5*). These findings suggest that the Netrin signaling microenvironment secreted by DNs is involved in regulating the horizontal projection of s-LNv axons.

### Ideas and speculation

The emergence of NetB-positive DNs and the aberrant axonal projection of s-LNv neurons caused by DN ablation occur concurrently during development (*Figure 6*). In *Drosophila*, three types of circadian oscillator neurons, including two DN1s, two DN2s, and four to five s-LNvs, can be identified as early as embryonic stage 16 (*Houl et al., 2008*). The DN3 group, which represents the largest contingent within the central circadian neuron network with over 35 neurons, emerges during the larval stage (*Liu et al., 2015*). Little is known about the function of DN3 neurons, particularly during the larval stage. Due to the lack of cell-specific labeling tools, we can only speculate about the identity of these newly generated Netrin-secreting neurons as DN3s based on their spatial distribution. Taken together, these results unveil a novel regulatory mechanism where axons, during the process of pathfinding, await guidance cues from newly born guidepost cells. This enables a switch in the direction of axon projection, facilitating the axon's subsequent journey.

# Materials and methods

**Key resources table**

| Reagent type (species) or resource | Designation | Source or reference | Identifiers | Additional information |
|---|---|---|---|---|
| Genetic reagent (*D. melanogaster*) | Pdf-GAL4 | Bloomington *Drosophila* Stock Center | RRID:BDSC_6900 | |
| Genetic reagent (*D. melanogaster*) | nSyb-GAL4 | Bloomington *Drosophila* Stock Center | RRID:BDSC_51941 | |
| Genetic reagent (*D. melanogaster*) | repo-GAL4 | Bloomington *Drosophila* Stock Center | RRID:BDSC-7415 | |
| Genetic reagent (*D. melanogaster*) | OK107-GAL4 | Kyoto Stock Center | DGRC-106098 | |
| Genetic reagent (*D. melanogaster*) | Tab2-201Y-G AL4 | Bloomington *Drosophila* Stock Center | RRID:BDSC-4440 | |
| Genetic reagent (*D. melanogaster*) | Tubulin-GAL4 | Bloomington *Drosophila* Stock Center | RRID:BDSC-5138 | |
| Genetic reagent (*D. melanogaster*) | per-GAL4 | Bloomington *Drosophila* Stock Center | RRID:BDSC-7127 | |
| Genetic reagent (*D. melanogaster*) | MB247-GAL4 | Bloomington *Drosophila* Stock Center | RRID:BDSC-50742 | |
| Genetic reagent (*D. melanogaster*) | Crz-GAL4 | Bloomington *Drosophila* Stock Center | RRID:BDSC-51976 | |
| Genetic reagent (*D. melanogaster*) | Pdf-GAL80 | Bloomington *Drosophila* Stock Center | RRID:BDSC-80940 | |
| Genetic reagent (*D. melanogaster*) | UAS--mCD8-GFP | Bloomington *Drosophila* Stock Center | RRID:BDSC-5137 | |
| Genetic reagent (*D. melanogaster*) | UAS--mCD8-GFP | Bloomington *Drosophila* Stock Center | RRID:BDSC-5130 | |
| Genetic reagent (*D. melanogaster*) | UAS--mCD8-RFP | Bloomington *Drosophila* Stock Center | RRID:BDSC-27392 | |
| Genetic reagent (*D. melanogaster*) | lexAop-mCD8-GFP | Bloomington *Drosophila* Stock Center | RRID:BDSC-32207 | |
| Genetic reagent (*D. melanogaster*) | Clk856-GAL4 | Bloomington *Drosophila* Stock Center | RRID:BDSC-93198 | |
| Genetic reagent (*D. melanogaster*) | UAS-rprC;;UAS-hid | Gifted from Yufeng Pan | N/A | |
| Genetic reagent (*D. melanogaster*) | Dscam1[1] | Bloomington *Drosophila* Stock Center | RRID:BDSC-5934 | |
| Genetic reagent (*D. melanogaster*) | Dscam1[21] | Gifted from Haihuai He | N/A | |
| Genetic reagent (*D. melanogaster*) | Dscam1[05518] | Bloomington *Drosophila* Stock Center | RRID:BDSC-11412 | |
| Genetic reagent (*D. melanogaster*) | NetA[Δ] | Bloomington *Drosophila* Stock Center | RRID:BDSC-66878 | |
| Genetic reagent (*D. melanogaster*) | NetB[Δ] | Bloomington *Drosophila* Stock Center | RRID:BDSC-66879 | |
| Genetic reagent (*D. melanogaster*) | NetAB[Δ] | Bloomington *Drosophila* Stock Center | RRID:BDSC-66877 | |
| Genetic reagent (*D. melanogaster*) | NetB-GFP | Bloomington *Drosophila* Stock Center | RRID:BDSC-67644 | |
| Genetic reagent (*D. melanogaster*) | NetB[tm] | Bloomington *Drosophila* Stock Center | RRID:BDSC-66880 | |

*Continued on next page*

*Continued*

| Reagent type (species) or resource | Designation | Source or reference | Identifiers | Additional information |
|---|---|---|---|---|
| Genetic reagent (*D. melanogaster*) | *UAS-Dscam1^RNAi* | Tsinghua Fly Center | THU3896 | |
| Genetic reagent (*D. melanogaster*) | *UAS-NetA^RNAi* | Tsinghua Fly Center | THU1972 | |
| Genetic reagent (*D. melanogaster*) | *UAS-NetB^RNAi* | Tsinghua Fly Center | TH201500623.S | |
| Genetic reagent (*D. melanogaster*) | *UAS-slit^RNAi* | Tsinghua Fly Center | THU1910 | |
| Genetic reagent (*D. melanogaster*) | *UAS-pak^RNAi* | Tsinghua Fly Center | TH201500668.S | |
| Genetic reagent (*D. melanogaster*) | *UAS-Dock^RNAi* | Tsinghua Fly Center | THU2815 | |
| Genetic reagent (*D. melanogaster*) | *UAS-SH3PX1^RNAi* | Tsinghua Fly Center | THU2738 | |
| Genetic reagent (*D. melanogaster*) | *UAS-tsr^RNAi* | Tsinghua Fly Center | THU0972 | |
| Genetic reagent (*D. melanogaster*) | *UAS-chic^RNAi* | Tsinghua Fly Center | THU0986 | |
| Antibody | anti-PDF (mouse monoclonal) | DSHB | C7; RRID:AB_760350 | IF(1:300) |
| Antibody | anti-HRP (rabbit monoclonal) | Jackson Immuno Research | Cat# 323-005-021, RRID:AB_2314648 | IF(1:500) |
| Antibody | anti-GFP FLUR 488 (rabbit polyclonal) | Invirtrogen | Cat# A-21311, RRID:AB_221477 | IF(1:200) |
| Antibody | anti-RFP (rabbit polyclonal) | Rockland | Cat# 600-401-379, RRID:AB_2209751 | IF(1:500) |
| Antibody | anti-GFP (chicken polyclonal) | Invirtrogen | Cat# A10262, RRID:AB_2534023 | IF(1:2000) |
| Antibody | anti-Myc (rabbit polyclonal) | Cell Signaling Technology | Cat# 9402, RRID:AB_2151827 | IF(1:200) |
| Antibody | Anti-Dscam1 18 mAb (mouse monoclonal) | Gift from Tzumin Lee (*Yu et al., 2009*) | N/A | IF(1:20) |
| Antibody | Goat Anti-Rabbit IgG H&L (Alexa Fluor 555) | Abcam | Cat# ab150078; RRID:AB_2722519 | IF(1:200) |
| Antibody | Goat Anti-Mouse IgG H&L (Alexa Fluor 647) | Abcam | Cat# ab150115, RRID:AB_2687948 | IF(1:200) |
| Antibody | Goat Anti-Rabbit IgG H&L (Alexa Fluor 488) preadsorbed | Abcam | Cat# ab150081; RRID:AB_2734747 | IF(1:200) |
| Antibody | Goat Anti-Mouse IgG H&L (Alexa Fluor 488) | Abcam | Cat# ab150113; RRID:AB_2576208 | IF(1:200) |
| Antibody | Goat Anti-Chicken IgY H&L (Alexa Fluor 488) | Abcam | Cat# ab150169; RRID:AB_2636803 | IF(1:200) |
| Recombinant DNA reagent | pUAST-HA-NetA (plasmid) | Gift from Duan R | N/A | |
| Recombinant DNA reagent | pUAST-HA-NetB (plasmid) | Gift from Duan R | N/A | |
| Software, algorithm | GraphPad Prism 8.0.2 | GraphPad | RRID:SCR_002798 | |
| Software, algorithm | Zeiss LSM Image Browser | Zeiss | https://www.zeiss.com/microscopy/int/downloads/lsm-5-series.html | |
| Software, algorithm | fiji | ImageJ | RRID:SCR_002285 | |

## Fly genetics

The flies were maintained on standard medium at 25°C with 60–80% relative humidity. The wild-type flies used in this study were $w^{1118}$. The *Dscam1 (CG17800)* null mutant allele, *Dscam1*$^{21}$, was obtained from Haihuai He's laboratory. *UAS-rpr.C;;UAS-hid* fly were kindly provided by Yufeng Pan's laboratory. The RNAi lines used in these studies were purchased from Tsinghua University; other flies were obtained from Bloomington Stock Center. Full genotypes of the flies shown in the main figures and figure supplements are listed in *Supplementary file 2*.

## Generation of transgenic flies

To generate the NetA and NetB transgenes, the full-length NetA cDNAs and the the full-length NetB cDNAs were subcloned into the pUAST vectors and injected into $w^{1118}$ flies.

## Antibodies

Antibodies were obtained from Developmental Studies Hybridoma Bank (anti-PDF, anti-FasII), Jackson Immuno Research (anti-HRP), Invitrogen (anti-GFP Alexa-488 goat anti-chicken IgY), Rockland (anti-RFP), and Abcam (Alexa-555 goat anti-rabbit IgG, Alexa-647 goat anti-mouse IgG, Alexa 488 goat anti-rabbit IgG, and Alexa 488 goat anti-mouse IgG), gifted from Tzumin Lee's Lab (anti-Dscam1).

## Collection embryo and larvae

~100 females lay eggs on the collecting medium (1% agar and 30% juice) in Petri dishes for a 4-hr period. To obtain larvae of defined ages, freshly emerged larvae were selected at 2-hr intervals. They were then allowed to grow on the cornmeal medium until they reached the desired age, and they were dissected at 0.5-hr intervals.

## Immunostaining

The embryos were dechorionated using 50% bleach and then fixed with 1 volume of heptane and 1 volume of 4% paraformaldehyde (PFA) for 20 min at room temperature. The vitelline membrane was removed by shaking the embryos in 100% methanol for 1 min. The embryos were washed with methanol for 5 min, followed by a 5-min wash with a 1:1 mixture of methanol and PBST (1× phosphate-buffered saline, PBS + 0.1% Triton X-100), and then washed twice with PBST for 5 min each. The embryos were blocked with 5% bovine serum albumin (BSA) in PBST for 30 min at room temperature. They were then treated with properly diluted primary antibody with 5% BSA in PBST at 4°C overnight. The embryos were washed three times with 1 ml PBST at room temperature for 30 min each. Subsequently, the embryos were treated with properly diluted fluorescent secondary antibody with 5% BSA in PBST and shaken at room temperature for 2–4 hr. The embryos were washed three times with 1 ml PBST at room temperature for 30 min each. After three washes, brains were mounted for microscopy in vectashield without 4',6-Diamidino-2'-phenylindole (DAPI) (Vector Laboratories).

Brain were dissected in PBS and fixed with 4% PFA for 25 min at room temperature. After fixation and three washes with 0.3% Triton X-100 in PBS. Brains were blocked with 5% goat serum in PBS with 0.3% Triton X-100 at 4°C 1 hr and were incubated with primary antibodies in PBS with 0.3% Triton X-100 at 4°C overnight. After four washes, brains were incubated with secondary antibodies in PBS with 0.3% Triton X-100 at room temperature for 2–3 hr. After four washes, brains were mounted for microscopy in vectashield without DAPI (Vector Laboratories).

Primary antibodies were mouse anti-PDF (1:300), mouse anti-FasII (1:50), Rabbit anti-HRP (1:500), mouse anti-Dscam1-18mAb (1:20), rabbit anti-GFP (1:200), rabbit anti-RFP (1:500), chicken anti-GFP (1:2000), rabbit anti-Myc (1:200), Secondary antibodies were Alexa Fluor 555 goat anti-Rabbit IgG (1:200), Alexa Fluor 647 goat anti-mouse IgG (1:200), Alexa Fluor 488 goat anti-rabbit IgG (1:200), and Alexa Fluor 488 goat anti-chicken IgY (1:200). Samples were imaged on an LSM 700 confocal microscope (Zeiss).

ImageJ software (National Institutes of Health) was used for the quantification vertical and horizontal axonal length of PDF immunostaining in s-LNvs. For s-LNvs horizontal measurements, results are presented as A.U. representing the fraction between the lengths of the horizontal projections divided by the distance between cell bodies and midline. Data are presented as means ± standard deviation (SD) from examined brains.

## RNAi screening

The UAS-RNAi line were obtained from Tsinghua Fly Center, and Bloomington *Drosophila* Stock Center. The RNAi screen fly was generated as follows: recombine *Pdf-GAL4* with *UAS-mCD8-GFP* or use *Clk856-GAL4* directly, which has a broader expression pattern. UAS-RNAi males were crossed to this two screening lines, and the resulting flies were kept at 25°C, and dissected at desired age.

## Neuron ablation

*Tab2-201Y-GAL4* which is expressed in larval MB γ neuron and *per-GAL4,Pdf-GAL80* which is expressed in DN neurons was used to drive the expression of reaper (rpr) and hid.The GAL4 flies integrated UAS-GFP, and the efficiency of ablation was confirmed by GFP signals in MB and DN neurons. Animals were raised at 25°C to desired time.

## Quantification and statistical analysis

All samples were obtained randomly and included in the statistical analysis. Statistical analysis was performed with GraphPad Prism V8.0.2 software. Data are presented as means ± SD. Shapiro–Wilk test was used to verify whether the data conformed to the normal distribution. $F$ test was used to verify homogenous variances between two groups. Bartlett test was used to verify homogenous variances three or more conditions and genotypes. Statistical significance was set as: $*p < 0.05$, $**p < 0.01$, $***p < 0.001$, $****p < 0.0001$, ns, no significance, $p > 0.05$. Statistical details of the experiments are found in the figure legends.

## Acknowledgements

We thank Dr. Haihuai He for providing *Dscam1* mutant flies; Dr. Yufeng Pan for the fly line used in cell ablation; Dr. Renjun Tu for the *NetB-GFP* fly; Dr. Xuan Guo for collecting the tool flies; the Bloomington stock center and Tsinghua fly center for providing flies; Dr. Tzumin Lee for Dscam1 antibodies; Dr. Ranhui Duan HA-NetA and HA-NetB plasmids. This work was supported by the STI2030-Major Projects (2021ZD0202500 to J.H.), the National Natural Science Foundation of China (32170970 to Y.T., 32230039 to J.H., and 32100788 to X.L.), and the Guangdong Key Project-2018B030335001 to J.H.

## Additional information

### Funding

| Funder | Grant reference number | Author |
|---|---|---|
| Ministry of Science and Technology of the People's Republic of China | STI2030-Major Projects 2021ZD0202500 | Junhai Han |
| National Natural Science Foundation of China | 32170970 | Yao Tian |
| National Natural Science Foundation of China | 32230039 | Junhai Han |
| National Natural Science Foundation of China | 32100788 | Xian Liu |
| Department of Science and Technology of Guangdong Province | Guangdong Key Project 2018B030335001 | Junhai Han |

The funders had no role in study design, data collection, and interpretation, or the decision to submit the work for publication.

### Author contributions

Jingjing Liu, Data curation, Formal analysis, Validation, Investigation, Visualization, Writing - original draft; Yuedong Wang, Investigation, Assisted with the RNAi screening; Xian Liu, Funding acquisition,

Investigation, Writing - review and editing; Junhai Han, Supervision, Funding acquisition, Project administration, Writing - review and editing; Yao Tian, Conceptualization, Supervision, Funding acquisition, Writing - original draft, Project administration, Writing - review and editing

### Author ORCIDs
Jingjing Liu ⓘ http://orcid.org/0009-0004-0780-2345
Junhai Han ⓘ https://orcid.org/0000-0001-8941-2578
Yao Tian ⓘ https://orcid.org/0000-0002-0367-6239

Reviewer #1 (Public Review): https://doi.org/10.7554/eLife.96041.3.sa1
Reviewer #2 (Public Review): https://doi.org/10.7554/eLife.96041.3.sa2
Author response https://doi.org/10.7554/eLife.96041.3.sa3

## Additional files

### Supplementary files
• Supplementary file 1. Full genotypes of the flies that are shown in the main figures and figure supplements.
• Supplementary file 2. List of genes used for RNA-interfering (RNAi) screen.
• MDAR checklist

### Data availability
The raw data has been included as supplements to the corresponding figures. Data from all experiments has been deposited at Dryad (https://doi.org/10.5061/dryad.gf1vhhmxj).

The following dataset was generated:

| Author(s) | Year | Dataset title | Dataset URL | Database and Identifier |
|---|---|---|---|---|
| Liu J, Wang Y, Liu X, Han J, Tian Y | 2024 | Data from: Spatiotemporal changes in Netrin/DSCAM1 signaling dictate axonal projection direction in Drosophila small ventral lateral clock neurons | https://datadryad.org/stash/share/cYlpTIyCWbJZf3mlfqLbCVrlETN5UWmn-HgFXLu6pT4 | Dryad Digital Repository, 10.5061/dryad.gf1vhhmxj |

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
