## [Editor Report · eLife assessment]

This study provides insights into the mechanism of axonal directional changes, utilizing the pacemaker neurons of the circadian clock, the sLNVs, as a model system. The data were collected and analysed using **solid** methodology, resulting in **valuable** data on the interplay of signalling pathways and the growth of the axon. The study holds potential interest for neurobiologists focusing on axonal growth and development.

---

## [Referee Report · Reviewer #1 (Public Review)]

The mechanisms of how axonal projections find their correct target requires the interplay of signalling pathways, and cell adhesion that act over short and long distances. The current study aims to use the small ventral lateral clock neurons (s-LNvs) of the *Drosophila* clock circuit as a model to study axon projections. These neurons are born during embryonic stages and are part of the core of the clock circuit in the larval brain. Moreover, these neurons are maintained through metamorphosis and become part of the adult clock circuit. The authors use the axon length by means of anti-Pdf antibody or Pdf>GFP as a read-out for the axonal length. Using ablation of the MB- the overall target region of the s-LNvs, the authors find defects in the projections. Next, by using Dscam mutants or knock-down they observe defects in the projections. Manipulations by the DNs - another group of clock neurons - can induce defects in the s-LNvs axonal form, suggesting an active role of these neurons in the morphology of the s-LNvs.

---

## [Referee Report · Reviewer #2 (Public Review)]

The paper from Liu et al shows a mechanism by which axons can change direction during development. They use the sLNv neurons as a model. They find that the appearance of a new group of neurons (DNs) during post-embryonic proliferation secretes netrins and repels horizontally towards the midline, the axonal tip of the LNvs. The experiments are well done and the results are conclusive.

---

## [Author Response]

The following is the authors’ response to the original reviews.

**Public Reviews:**

**Reviewer #1 (Public Review):**
Summary:The mechanisms of how axonal projections find their correct target requires the interplay of signalling pathways, and cell adhesion that act over short and long distances. The current study aims to use the small ventral lateral clock neurons (s-LNvs) of the *Drosophila* clock circuit as a model to study axon projections. These neurons are born during embryonic stages and are part of the core of the clock circuit in the larval brain. Moreover, these neurons are maintained through metamorphosis and become part of the adult clock circuit. The authors use the axon length by means of anti-Pdf antibody or Pdf>GFP as a read-out for the axonal length. Using ablation of the MB- the overall target region of the s-LNvs, the authors find defects in the projections. Next, by using Dscam mutants or knock-down they observe defects in the projections. Manipulations by the DNs - another group of clock neurons- can induce defects in the s-LNvs axonal form, suggesting an active role of these neurons in the morphology of the s-LNvs.Strengths:The use of *Drosophila* genetics and a specific neural type allows targeted manipulations with high precision.Proposing a new model for a small group of neurons for axonal projections allows us to explore the mechanism with high precision.Weaknesses:It is unclear how far the proposed model can be seen as developmental.The study of changes in fully differentiated and functioning neurons may affect the interpretation of the findings.

We appreciate the reviewer's feedback on the strengths and weaknesses of our study.

We acknowledge the strengths of our research, particularly the precision afforded by using *Drosophila* genetics and a specific neural type for targeted manipulations, as well as the proposal of a new model for studying axonal projections in a small group of neurons.

We understand the concerns about the developmental aspects of our proposed model and the use of Pdf-GAL4 >GFP as a read-out for the axonal length (revised manuscript Figure 1--figure supplement 1). However, even with the use of Clk856-GAL4 that began to be expressed at the embryonic stage (revised manuscript Figure 3--figure supplement 1) to suppress Dscam expression, the initial segment of the dorsal projection of s-LNvs (the vertical part) remained unaffected. Instead, the projection distance is severely shortened towards the midline, and this defect persists until the adult stage. It is for this reason that we delineate the dorsal projections of s-LNvs into two distinct phases: the vertical and horizontal parts, rather than a mere expansion in correspondence with the development of the larval brain.

Thank you for your valuable feedback, and we have incorporated these considerations into our revised manuscript to enhance the clarity and depth of our research.

**Reviewer #2 (Public Review):**
Summary:The paper from Li et al shows a mechanism by which axons can change direction during development. They use the sLNv neurons as a model. They find that the appearance of a new group of neurons (DNs) during post-embryonic proliferation secretes netrins and repels horizontally towards the midline, the axonal tip of the LNvs.Strengths:The experiments are well done and the results are conclusive.Weaknesses:The novelty of the study is overstated, and the background is understated. Both things need to be revised.

We appreciate your acknowledgment that the experiments were well-executed and the results conclusive. This validation reinforces the robustness of our findings.

We take note of your feedback regarding the novelty of the study being overstated and the background being understated. While axonal projections navigate without distinct landmarks, like the midline or the layers, columns, and segments, they pose more challenges and uncertainties. As highlighted, our key contribution lies in elucidating how axonal projections without clear landmarks are guided, with our research demonstrating how a newly formed cluster of cells at a specific time and location provides the necessary guidance cues for axons.

We value your insights, and we have carefully addressed these points in our manuscript revision to improve the overall quality and presentation of our research.

**Recommendations For The Authors:**

**Reviewer #1 (Recommendations For The Authors):**
The overall idea of using the s-LNvs as a model is indeed intriguing. There are genetic tools available to tackle these cells with great precision.However, based on the stage at which these cells are investigated raises some issues, that I feel are critical to be addressed.These neurons develop their axonal projections during embryogenesis and are fully functioning when the larvae hatch, thus to investigate axonal pathfinding one would have to address embryonic development.The larval brain indeed continues to grow during larval life, however extensive work from the Hartenstein lab, Truman lab, and others have shown that the secondary (larval born) neurons do not yet wire into the brain, but stall their axonal projections.It is thus quite unclear, what the authors are actually studying.One interpretation could be that the authors observe changes in axon length due to morphological changes in the brain. Indeed, the fact that the MB expands the anatomy of the surrounding neuropil changes too.Moreover, it is unclear when exactly the Pdf-Gal4 (and other drivers) are active, thus how far (embryonic) development of s-LNvs is affected, or if it's all happening in the differentiated, functioning neuron. (Gal4 temporal delay and dynamics during embryonic development may further complicate the issue). As far as I am aware the MB drivers might already be active during embryonic stages.Since the raised issue is quite fundamental, I am not sure what might be the best and most productive fashion to address this.Eg. either to completely re-focus the topic on "neural morphology maintenance" or to study the actual development of these cells.

We thank the reviewer for the detailed and insightful feedback on our study. We have tested whether Pdf-Gal4 could effectively label s-LNv, and tracked the s-LNv projection in the early stage after larvae hatching. We did not observe the PDF antibody staining signal and the GFP signal driven by Pdf-GAL4 when the larvae were newly hatched. At 2-4 hours ALH, PDF signals were primarily concentrated at the end of axons, while GFP signals were mainly concentrated at the cell body. Helfrich-Förster initially detected immunoreactivity for PDF in the brains approximately 4-5 hours ALH. The GFP signal expressed by Pdf-GAL4 driver does have signal delay. However, at 8 hours ALH, the GFP signal strongly co-localized with the PDF signal within the axons (see revised manuscript lines 98-101) (Figure 1—figure supplement 1).

Based on previous research findings and our staining of Clk856-GAL4 >GFP, it is indeed confirmed that the dorsal projection of s-LNvs in *Drosophila* is formed during the embryonic stage (Figure 3—figure supplement 1). The s-LNvs in first-instar larval *Drosophila* are capable of detecting signal output and may play a role in regulating certain behaviors. Our selection of tools for characterizing the projection pattern of s-LNv was not optimal, leading us to overlook the crucial detail that the projection had already formed during its embryonic stage.

However, even when employing Clk856-GAL4 to suppress Dscam expression from the embryonic stage, the initial segment of the dorsal projection of s-LNvs (the vertical part) remains unaffected. Instead, the projection distance is severely shortened towards the midline, and this defect persists until the adult stage. It is for this reason that we delineate the dorsal projections of s-LNvs into two distinct phases: the vertical and horizontal parts, rather than a mere expansion in correspondence with the development of the larval brain.

From the results searched in the Virtual Fly Brain (VFB) database (https://www.virtualflybrain.org/), it is clear that the neurons that form synaptic connections with s-LNvs at the adult stage are essentially completely different from the neurons that are associated with them at the L1 larval stage. Thus, most neurons that form synapses with s-LNvs in the early larvae either cease to exist after metamorphosis or assume other roles in the adult stage. Similar to the scenario where Cajal-Retzius cells and GABAergic interneurons establish transient synaptic connections with entorhinal axons and commissural axons, respectively, these cells form a transient circuit with presynaptic targets and subsequently undergo cell death during development. In our model, the neurons that synapse with s-LNvs in early development serve as "placeholders," offering positive or negative cues to guide the axonal targeting of s-LNvs towards their ultimate destination.

Thank you again for your valuable feedback, and we have incorporated these considerations into our revised manuscript to enhance the clarity and depth of our research.

**Reviewer #2 (Recommendations For The Authors):**
Major:In the introduction too many revisions are cited and very few actual research papers. This should be corrected and the most significant papers in the field should be cited. For example, there is no reference to the pioneering work from the Christine Holt lab or the first paper looking at axon guidance and guideposts by Klose and Bentley, Isbister et al 1999.The introduction should encapsulate the actual knowledge based on actual research papers.

We acknowledge your concern regarding the citation of review papers rather than primary research papers in the introduction. Following your suggestion, we have revised the introduction section to incorporate references to relevant research papers.

In the introduction and discussion: The authors cite revisions where the signals that guide axons across different regions including turning are shown and they end up saying: "However, how the axons change their projection direction without well-defined landmarks is still unclear." I think the sentence should be changed. Many things are still not clear but this is not a good phrasing. Maybe they could focus on their temporal finding?

We appreciate the reviewer's feedback and insightful suggestions. We agree that emphasizing the temporal aspect is crucial in our study. However, we also recognize the significance of understanding the origin of signals that guide axonal reorientation at specific locations. While axonal projections navigating without distinct landmarks pose more challenges and uncertainties compared to those guided by prominent landmarks like the midline, our research demonstrates the crucial role of a specific cell population near turning points in providing accurate guidance cues to ensure precise axonal reorientation. We have revised our phrasing in the introduction and discussion to better reflect these key points (see revised manuscript lines 69-71 and 350-354). Thank you for highlighting the significance of focusing on our temporal findings and the complexities involved in studying axonal projection.

Many rather old papers have looked into the effect of repulsive guideposts to guide axon projections. In particular, I can think of the paper from Isbister et al. 1999 (DOI: 10.1242/dev.126.9.2007) that not only shows how semaphoring guides Ti axon projection but also shows how the pattern of expression of sema 2a changes during development to guide the correct projection. I really think that the novelty of the paper should be revised in light of the actual knowledge in the field.

We appreciate the reviewer's reference to the seminal work by Isbister et al. (1999) and the importance of guidepost cells in axon projection guidance, which we have already cited in our revised manuscript. It is crucial to recognize that segmented patterns such as the limb segment traversed by Ti1 neuron projections or neural circuits formed in a layer- or column-specific manner also serve as intrinsic "guideposts," offering valuable insights into axonal pathfinding processes. In our model, explicit guidance cues are lacking. As highlighted, our key contribution lies in elucidating how axonal projections without clear landmarks are guided, with our research demonstrating how a newly formed cluster of cells at a specific time and location provides the necessary guidance cues for axons (see revised manuscript lines 350-354). We have ensured that our revised manuscript reflects these insights and emphasizes the significance of studying axonal guidance in the absence of distinct guideposts. Thank you for underscoring these essential points, which enhance our understanding of axonal projection dynamics.

Minors:Line 54, the authors start talking about floorplate at the end of a section on *Drosophila*. Please use “In vertebrates”, or “in invertebrates” or “in *Drosophila*” etc.. when needed to put things in context.

We thank the reviewer for this suggestion and have modified this sentence. Please refer to lines 62-63 of the revised manuscript.

Line 69: many factors change the axonal outgrowth. The authors are missing the paper from Fernandez et al. 2020, who have shown that unc5 the receptor of netrin induces the stalling for sLNvs projections before the turn. https://doi.org/10.1016/j.cub.2020.04.025

We thank the reviewer for this suggestion and have added this research article. Please refer to line 79 of the revised manuscript.

Line 99: "precisely at the pivotal juncture". It I hard to see how it was done in the figures shown. Can the authors add a small panel with neuronal staining showing this (please no HRP)?For all figures, tee magenta is too strong and it is really hard to see the sLNvs projections. Can this be sorted, please?

We have depicted the pivotal juncture in the schematic diagram on the left side of Figure 1C. Additionally, we have included a separate column of images without HRP in Figure 1A. Moreover, we have modified the pseudo-color of HRP from magenta to blue to enhance the visualization of the s-LNv projection. The figure legends have also been correspondingly modified.

Line 407: Spatial position relationship between calyx and s-LNvs. OK107-GAL4 labels ... calyx and s-LNvs labeled by, which which.

We have modified it according to your suggestion. Please refer to lines 430-432 of the revised manuscript.

Line 137 typo RPRC

We thank the reviewer for noticing this mistake, which has now been corrected. Please refer to line 148-149 of the revised manuscript.

Section 158-164. the paper from Zhang et al 2019 needs to be cited since they have found the same effect of decreasing Dscam even if they didn't think about horizontal projection.

Thanks to the suggestion, we have included in the manuscript the phenotype observed by Zhang et al. (2019) upon knocking down Dscam1-L in adults. Please refer to lines 170-172 of the revised manuscript.

Line 176: typo senses (instead of sensor).

Thank you for pointing out our mistake. We have modified it according to your suggestion. Please refer to line 189 of the revised manuscript.

Line 193: more than Interesting it is Notable. Add "ubiquitus" knockdown.

Thank you for the suggestion. We have included the word "ubiquitus" to enhance the precision of the narrative. Please refer to line 206 of the revised manuscript.

Line 224: the pattern of expression of the crz cells is not visible where the projections of sLNvs are located. Are they in that region? Or further away?

We've changed the pseudo-color of HRP, and in the updated Figure 5- figure supplement 1, you can see the projection pattern of crz+ cells, positioned close to the end of the s-LNv axon terminal.

Line 243: applied? Do you mean "used"

Thank you for the suggestion. We have revised it at line 256.

Figure 5 Sup1: the schematic shows DNs proliferation that is not visible on the GFP image. Please comment.

We have modified the Figure 5 figure supplementary 1 for 120 h per-GAL4, Pdf-GAL80 >GFP expression pattern. Due to the strong GFP intensity in some DN neurons, there was a loss of GFP signal. Additionally, in Figure 6 figure supplementary 1, we have added co-localization images of DN and s-LNv at 72 h and 96 h. To better illustrate the co-localization information, we have shown only a portion of the layers in the right panel. We hope these additions clarify your concerns.

Line 251: cite Fernandez et al. 2020 with Purohit et al 2012.

We have modified it according to your suggestion. Please refer to line 264 of the revised manuscript.

Line 272: you have not shown synergistic effects because you have not modulated both pathways at the same time. You should talk about complementary.

We have modified it according to your suggestion at lines 25, 285, 439.